# Medicinal Plants in Semi-Natural Grasslands: Impact of Management

**DOI:** 10.3390/plants11030353

**Published:** 2022-01-27

**Authors:** Marika Kose, Indrek Melts, Katrin Heinsoo

**Affiliations:** Chair of Biodiversity and Nature Tourism, Estonian University of Life Sciences, Kreutzwaldi 5, 51007 Tartu, Estonia; marika.kose@emu.ee (M.K.); indrek.melts@emu.ee (I.M.)

**Keywords:** biodiversity, ecosystem services, grassland, herbal medicine, land-use, nature conservation, plant ecology, sustainable management

## Abstract

Semi-natural grasslands (SNG) are valuable for their high biodiversity, cultural and landscape values. Quantitative information about medicinal plants (MP) in SNG facilitates the evaluation of ecosystem services of these habitats. Different literature sources were used to assess the ratio and frequency of MP species in several Estonian SNG and to evaluate the impact of management on these values. Lists of MP species according to different MP definition scenarios are available. The ratio of MP species in the local plant species list was the largest in alvars, followed by floodplain and wooded meadows. The average number of MP species in wooded meadows and alvars was about twice of that found in naturally growing broadleaved forest (according to the most detailed MP species list, 7.2, 7.8 and 4.3 plot^−1^, respectively). Fertilization of wooded meadows had no significant impact on MP species ratio, but decreased the percentage of MP biomass. Coastal meadows had few MP species and the impact of management quality depended on adopted MP scenarios. Comparison of Ellenberg indicator values revealed that MP species were more drought-tolerant, with higher commonness and more anthropophyte than the rest of studied grassland species.

## 1. Introduction

Semi-natural grasslands (SNG) as valuable ecosystems have been in focus for several decades due to their high biodiversity, cultural importance and landscape values. Their restoration has been challenging, but this activity has also led to improved knowledge about and opportunities for these unique ecosystems [1,2]. Typically, biomass production of these areas is lower than that of intensively managed grass and croplands due to restrictions on ploughing, sowing and fertilization and, therefore, to make extensive management an attractive option for landowners, economic stimulation measures should be adopted. In the EU, this problem is partly solved through the agricultural subsidy schemes for NATURA 2000 network areas that should compensate the loss of crop production due to extensive management. Valuable grasslands, however, are not determined by this network only and are found in other locations and geographical regions as well [3,4]. Recent approaches to evaluation of SNG by the complex list of potential ecosystem services (ES) provides a much more holistic overview and supports sustainable development ideas in different regions by giving a valuable input to the benefits communities can get and share through different provider groups [5,6]. Different authors point out the wide range of goods and services provided by grasslands, but beside plant species diversity in general, all of these assume that SNG are valuable refuges for pollinators [2], sources of medicinal plants [3,6] and habitats for migratory and breeding birds [7].

Compared with SNG biodiversity or pollinator richness, the occurrence, diversity and dynamics of medicinal plants (MP) is less studied. The lack of scientific knowledge about the potential health benefits is an issue while the phytochemical content and pharmacological actions in order to define efficacy and safety have been studied for a limited number of species, but this situation may change rapidly [8]. Hitherto, available ethnobotanical reviews focus mainly on the regions where the old traditions and indigenous knowledge have survived globalization tendencies in different parts of the world: Africa [4], Europe [6,9], Asia [10].

Application of these databases to any ecological study is disputable as in most cases, the native names used in local folklore are difficult to match with scientific plant names in Latin. Moreover, the historic records also contain MP species which are no more suggestable due to their severe negative side effects and, therefore, the scientific definition of MP species has been exploited mainly in anthropological context so far and its combination with ecological context needs to be revealed. On the other hand, there are some plant species that are internationally recognized as MP and, therefore, studied in detail for biorefinery purposes. This information allows us to make some assumptions about the environmental factors that may have an impact on the potential of SNG as a source of medicinal plants.

There are several reports available, demonstrating that medicinal plants take up and accumulate metals from the growing substrate [11,12] and, hence, the unfertilized, unplowed SNG may provide plant biomass with higher quality than from traditional agricultural fields where mineral fertilizers are used. The production of compounds that are typically associated with healing benefits are secondary metabolites (alkaloids, glycosides, polyphenols and terpenes); however, it is highly dependent on local environment conditions and, therefore, moderate abiotic stress is assumed to be useful for pharmaceutical purposes [13]. For instance, it has been reported that the monoterpene concentration in *Salvia officinalis* increased significantly during drought stress [14] and that *Glycyrrhiza uralensis* roots contained more useful compounds when grown under a low light intensity regime [15]. Other authors have assumed that the healing, antimicrobial and antitumor effects of MP can be associated with secondary metabolites that protect these plants against free radicals and prevent photosynthetic process damage and demonstrated with their meta-analyses that water scarcity is only one possible factor increasing the ratio of phenolic compounds [16]. The list of environmental factors that may have an impact on useful chemical production also includes soil nutrients and the local agro-climate [17,18]. Besides abiotic factors, the impact of neighboring plants should be also considered for MP quality, and MP frequently have a rich rhizosphere community consisting of different arbuscular mycorrhizal fungi genera [19]. Mutualistic interaction has been demonstrated to increase the concentration of phenolic acids in the roots of model MP *Arnica montana* [20]. Moreover, different endophytic fungi may be linked to the therapeutic activity of the *Asteraceae* family in different parts of the world, but without a host plant they seem to be inactive [21]. Hence, it can be assumed that MP that grow naturally in SNG with high biodiversity may provide us with valuable healing compounds. This is already recognized in practice, farmers value species-rich grasslands for these positive effects on livestock health [22]. However, data on the ratio of MP species or their frequency in any European SNG type are limited and, consequentially, information about the regional potential of MP species is difficult to gather. Such a shortage of knowledge also limits any attempt to quantifying the level of ecosystem services provided by SNG.

In order to fill the gap, we endeavored to find any publications with the lists of MP species that were available for the areas where we had previously performed our studies about the impact of management on SNG plant biodiversity [23,24]. Fortunately, we were able to detect three sources for Estonian conditions that have been published for different purposes. Simultaneous applicability of each of them in different scenarios ensured that we consider different aspects of MP species concept and are more available to analyze the dynamics of this particular ecosystem service in its broader scale.

The main aim of the current study is to quantify the potential of MP as an ecosystem service provided by SNG. For this purpose, we studied the ratio and frequency of MP species in different SNG types, modelling their availability according to species accumulation curves [25], compared these values with those of natural habitat types and analyzed the impact of different management options (fertilization, mowing) on these values. The database of Estonian SNG herbaceous plants has been supplemented with the traits and indicator values of species’ environmental requirements (details in [26]), from this data, we analyzed whether MP species have any particular environmental preferences compared with other SNG plant species. 

## 2. Results

Compiling different floristic lists of Estonian habitat types resulted in 538 plant species and a 10,814-record database that was used for further analyses. The largest list of plant species was obtained from wooded meadows (330) while in the second most diverse SNG type (floodplain meadows) one third fewer plant species (194) was recorded (Table 1). The species accumulation curves demonstrated that deceasing the amount of study sites to 50% could diminish the plant species lists by half (from 57% in heaths to 28% in wooded meadows) (Table 2, original species accumulation curves Appendix A). The MP availability in this case lessened in the same range depending on the applied scenario.

The ratio of MP species in the total species list depended on the scenario; the biggest differences were detected for the species list of heaths, which was the type with one of the poorest MP species potential (% of MP species in total species list) according to scenario 1, but the best potential according to scenario 3 (Table 1). A similar pattern was found in the analysis of floristic databases of different type inventories where almost half the recorded plants in different heaths belonged to scenario 3 MP list. In general MP% was higher in natural habitats and among SNG the alvars had the largest relative potential of MP species followed by floodplain and wooded meadows (Figure 1A). Frequency of MP species according to most scenarios was the highest in wooded meadows and alvars followed by broad-leaved forests. In coastal meadows, where the absolute numbers according to species list (Table 1) was comparable to that of alvars, the frequency of MP (MPfreq) was much lower. Among SNG the lowest MPfreq was detected in fens, but that was higher than the values in heaths despite the scenario (Figure 1B). The other SNG type where MP species were less common, were floodplain meadows.

The analyses of the Laelatu wooded meadow fertilization experiment database revealed that according to two of the three scenarios, MP% increased in time. The impact of fertilization on MP% was minimal and most of the MP species survived the additional nutrient application indicating adaptation and strain of these species (Figure 2). At the same time, major differences in the percentage of MP biomass (MPb%) between fertilization treatments occurred; MPb% during N fertilization decreased significantly and according to scenarios 1 and 3, these differences in the treatment with highest additional nutrient application (NPK2) have not reversed even 20 years after the end of the experiment (Figure 2D,F). 

The coastal meadows database was difficult to study as among Estonian SNG they seem to be one of the poorest SNG types in terms of MP species numbers (Table 1). Hence, application of different scenarios caused significant differences in the evaluation of coastal meadows potential in terms of MP species growing type. Good management of coastal meadows tends to decrease the MP ratio in the inventory species list (Figure 3A). The impact of management of MP species frequency is species-specific and may result in contrasting impacts (Figure 3B).

With the principal component analyses of SNG species, we could not identify any clear peculiarities for MP species. Due to gaps in our knowledge about some characteristics, only 171 species were included in the analyses and despite the scenario, the first two axes did not explain more than 35% of the variation. According to the different scenarios, the best correlation was found between MP and its commonness—MP species are more likely available in nature than the rest of the grassland species (Figure 4). They also tend to be less light-demanding and tolerate drier growing areas. The average and maximum theoretical height does not define MP species; moreover, they do not differ from the rest of the grassland species by their N demand. 

## 3. Discussion

In general the plant species accumulation curves in our study did not reach any plateau to describe the asymptotic richness [25]. The exceptions were broad-leaved forest and fen habitats, which were either natural or long-time abandoned and, therefore, turned to natural succsession. All SNG demonstrated the potential for more species, if there were more plant community species lists available. Most probably, such high variability was caused by very different managemant background and quality of our study sites. In our previous papers, we have demonstrated that both restoration quality and management acitivities have a very strong impact on local plant biodiversity [23,24]. If we deminished our study site selecion in each habitat type by half, almost half of the biodiversity of habitat type could be out of our database, and we recorded almost twice less MP species. However, the maximum plant species numbers per habitat were in the range of previous studies of these habitats and, therefore, we assumed our database to be representative [27].

The longest list of plant species and also the largest number of MP species was obtained for wooded meadow type. This result is in agreement with other reports showing wooded meadows to be the SNG with the largest plant species richness on a small scale [28]. Such a phenomenon can be partly explained by the large variability of environmental conditions inside this particular type and amplified in our study by the inclusion of data from wooded meadow sites with widely differing water availability and tree coverage. The ratio of MP (MP%) in wooded meadows, however, was modest and can indicate both large ratio of sedge species and grasses with no recognized healing value in Estonia, or the existence of rare species which usage whose medicinal value has not been recorded in any of our scenarios. The ranking by MP species in total type plant species list varied by scenario, but dry types (alvar) included proportionally less MP species than types which are at least partly seasonally flooded or are permanently water-saturated and, therefore, higher local diversity of environmental factors are expected (floodplain meadows). Natural habitat types tended to have a relatively higher number of MP species, but the comparatively short species list in heaths make its position in the ranking list extremely influenced by scenario used. 

The average MP% values by types follow a largely similar pattern; the largest percentage of MP species was recorded from natural habitat types and almost half of the heath plant species were mentioned in the World Encyclopedia of Medicinal Plants [29]. Among SNG, however, alvars were the preferred sites according to this criterion, indicating that the MP species in total type list are found in various sites. Wooded and floodplain meadows demonstrated similar MP% despite applied scenario. The smallest MP% was found for fens indicating that the MP species list there included more rare species that are not growing in each studied site. One can also speculate that fens are historic SNG that are difficult to manage due to high groundwater levels found there [30] and, therefore, the dataset could contain sites with different edaphic conditions and/or management history and, therefore, variable plant species lists. 

The average number of MP per plot (MPfreq) was the largest in wooded meadows and alvars and, therefore, these plant community types should be preferred regarding MP provision ecosystem service. The quantity of MP species to be found in plots of these types is most probably twice than that of natural broad-leaved forest, which is the future of neglected wooded meadow in the absence of a proper traditional management regime. Despite the scenario, MPfreq in coastal and floodplain meadows was about half that of preferred sites with more moderate water availability. During their adaptation to water deficit the species growing in alvars and wooded meadows may have increased the production of secondary metabolites that are associated with the pharmaceutic impact [13,14].

A long-term study of Laelatu wooded meadow demonstrated that continuous management of SNG can be rewarded by additional MP species emergence in the site—in our case, even some species from the strictest scenario 3 started to grow in the area. Fertilization of the plots decreased the overall number of species on plots drastically [23]. According to our calculations, however, MP% changes between control/fertilization during the experiment were insignificant, indicating MP species to be vital and as tolerant of environmental change as the rest of the plant community. On the other hand, the aim of the annual fertilization in this experiment was to increase total biomass yield, and that target was achieved by between a 250 and 400 % greater annual plant biomass production per area [23]. The large decrease of MPb% of fertilized plots compared with control ones (NPK1 and NPK2 despite scenario) reveals that other plants than MP have caused the production increase. This indicates that MP, growing in control plots, did not suffer from N deficit and, therefore, N application is not required to increase their yield for economic reasons. 

Our previous study has revealed that in coastal meadows plant species, number does not increase with long-term management [24] and in comparison with other types studied, only a small number of MP species were present here. Therefore, the interpretation of obtained results must be done carefully. A high percentage of MP in sites that are managed for shorter periods indicate that the natural habitat types developing from coastal meadows during natural succession process have longer list of MP species. Moreover, the scenario-dependent effect of management quality on MP frequency in this type can be a result of the dominance of different plant species in groups compared. For instance, MP species list in scenario 3 includes Phragmites australis, which is used as an indicator species for detecting poor management of coastal meadows. In scenario 2, on the other hand, MP species list contained more species, which was probably favored by proper meadow management.

This holistic approach to studying MP plant and environment characteristics resulted in PCAs that explained one third of the class variability. According to all the scenarios used, the MP species tolerate or prefer dry growing conditions. Such a result is in accordance with our results which showed a greater frequency of MP species in alvars and wooded meadows as compared with floodplain meadows. Both water shortage and shade tolerance of MP species can be linked to the increased synthesis of useful secondary metabolites during environmental stress [31,32]. MP species also tend to be more common and anthropophyte than the other plant species in the community. The current analysis was only carried out on less than 200 grassland species; however, and for a better understanding of the ecological and social patterns of MP species, the database should be developed to include all the herbaceous plant species of Estonia regardless of their growing type. Indeed, the trends observed can also be related to historical factors; during the country’s oral heritage period, common plants around homesteads had a greater probability of being used and the knowledge about their healing properties being transferred to the next generation. If so, one might speculate that the large biodiversity in Estonian NATURA 2000 sites and the currently improving floristic knowledge might lead to a major increase in the wider MP potential of SNG. For instance, in Estonia there is no evidence of use of sedges as MP species, but there is proof from other regions with similar climates that they can have a positive impact on our health [33,34,35].

In conclusion we find that the MP potential of grasslands today is significantly dependent on cultural traditions and both the available MP species list and local socio-economic situation should be considered while giving a quantified evaluation to this particular ecosystem service in a region or ecosystem type. The risk of MP contamination with zoonotic disease during the grazing period should also be clarified [36] and SNG management plans adopted which guarantee consumer safety. In addition, there are reports which demonstrate evidence of overexploitation of natural MP resources [37], and therefore both harvest pressure and economic feasibility of MP utilization for local community and landowners should be assessed through further close cooperation of ecologists, economists and pharmacists. 

## 4. Materials and Methods

### 4.1. Medicinal Plant Species List 

We composed the Estonian list of MP species on the basis of three different sources that were created and published for different purposes [29,38,39]. These publications included very different plant species and therefore the data from each particular source was the basis of a separate scenario in each implicated analysis (Table 3, full lists of MP species by scenarios are available in Appendix A). The current analyses involve herbaceous and low-height shrub plant species (hereby “plants”). Trees and bushes were excluded from the study as their abundance in SNG usually indicates poor management and, therefore, a negligible provision of ecosystem services from those areas. Low-height shrubs that tolerate periodic haymaking or grazing were included in the MP species lists as they could include widely popular MP species (e.g., Oxycoccus palustris in fens).

### 4.2. Floristic Databases and Created Parameters

We gathered the information about plant species and their frequency in various SNG from databases that had been collected and explored in a range of studies undertaken for specific purposes. To determine the occurrence and frequency of MP species in various types, we scanned the database that included inventory results from 1999–2019 in 82 sites of 12 different NATURA 2000 network habitats (Table 4). The majority of the plant species lists from the plant communities (one site could include several plant communities with different management history or study year) was extracted from the national environmental monitoring information system managed by the Estonian Environment Agency who initiates biodiversity inventories across a range of valuable habitats, particularly in NATURA 2000 areas. Data from heaths and old hemi-boreal broad-leaved forests were also included in the study as references in order to evaluate the MP potential of SNG. The floristic database was not balanced; some habitats had been visited more frequently than others and the timespan of inventories varied from unique to multiple visits. However, none of the particular study sites was was particularly overwhelming in the habitat and incorporating the habitat data into larger types improved the database quality (Table 4). The minimal potential percentage of species in half sites per habit type was calculated as ratio of median and maximum species numbers per habit type. The Shannon E index was used to evaluate the evenness of the habitat types. 

The percentage of MP (MP%) was calculated as the ratio between the sums of MP species by particular scenario (MP) and total number of plant species in the sites’ (s) plant species lists of all sites of particular habitat type (TP) multiplied by 100 to get %:MP% = 100 ∗ ∑MP_s_/∑TP_s_.(1)

With such a detailed calculation method, we tried to minimize the impact of one occasional record of any rare species/untypical site and generalize the data obtained from different study sites.

The size of inventory plots in the database varied from 0.2 × 0.2 to 1 × 1 m depending on study year/location and the number of plots per site was from 10 to 80 (Table 2). In each plot, the occurrence of particular species had been recorded without any additional data regarding its abundance in the plot. Hence, we were not able to detect the abundance of MP in the site/type, only the probability to find these from a plot of a particular type. The number of plots studied per any type was large (between 280 and 2800) and, therefore, we assume this probability to be close to the MP occurrence frequency and use this term as more understandable for ecologists than the “number of MP species per plot”. In practice the average MP frequency (MPfreq) in any type was calculated as the ratio between the sum of the plots per site(s) where this particular MP species was recorded (MPplot) divided by the total number of plots inventoried in this particular type (plottot):MPfreq = ∑MPplot_MPs_/plot_tot_.(2)

For the analysis on the impact of fertilization on MP species performance, we used the dataset of Laelatu wooded meadow, western Estonia, which has been mowed annually for approximately 300 years [26]. The fertilization experiment in this meadow took place between 1961 and 1981 with annual fertilization of PK and two different levels of N in three plots per treatment (details in [29]). In order to avoid the annual effect of weather and to minimize the dynamics of vegetation response to treatment change, we compared only two periods in our current analyses: the period in which we assume the impact of fertilization was steady (1969–1981) and the period where no further impact of fertilization on biomass production was detected (2005–2016). The dataset allowed us to calculate the ratio of MP biomass (MPb%) per treatment as the sum of the MP dry biomass weights (w) measured in particular treatment plots divided by the sum of all species (tot) dry biomass in the same plots:MPb% = 100 ∗ ∑w_MPp_/∑w_totp_.(3)

The case study on management impact was performed on the same data about coastal meadows that were added to the database for SNG types comparison. These data originated from a study of 14 coastal meadows, western Estonian coastal meadows (https://doi.org/10.15159/eds.dt.21.01, 25 January 2022). Most of the sites had been annually grazed by cattle and the management varied from permanently managed to recently restored. The division of these sites into three groups based on their management history and vegetation quality has been proved by their plant community characteristics [24] and we, therefore, continued the current study with the same “very good”, “good” and “poor” management (vegetation quality) classes.

### 4.3. Statistical Analyses

Lists of MP species depended significantly on the implicated scenario and, therefore, no further statistical analysis was reasonable for evaluating the impact of different management regimes on MP species occurrence. We compared the general characteristics of MP species and the other SNG plant species with the CANOCO software unconstrained principal component analysis. The not-compositional response data included different Ellenberg values for each species if available (classes indexed into numbers according to [26], details in Table 5) and theoretical average and maximum heights obtained from the literature [41].

## 5. Conclusions

The ratio of MP species in the local total plant species list was the largest in alvars, followed by floodplain and wooded meadows.The average number of MP species per study site in wooded meadows and alvars was about twice that found in naturally growing broad-leaved forest (according to the most detailed MP species list, 7.2, 7.8 and 4.3, respectively).Fertilization did not decrease MP species ratio in Estonian wooded meadow, but decreased the percentage of MP biomass in total yield.The frequency of MP species in Estonian coastal meadows depended on applied MP definition scenario.Principal component analysis revealed that MP species are more drought-tolerant, with higher commonness and more anthropophyte than the rest of studied grassland species.

## Figures and Tables

**Figure 1 plants-11-00353-f001:**
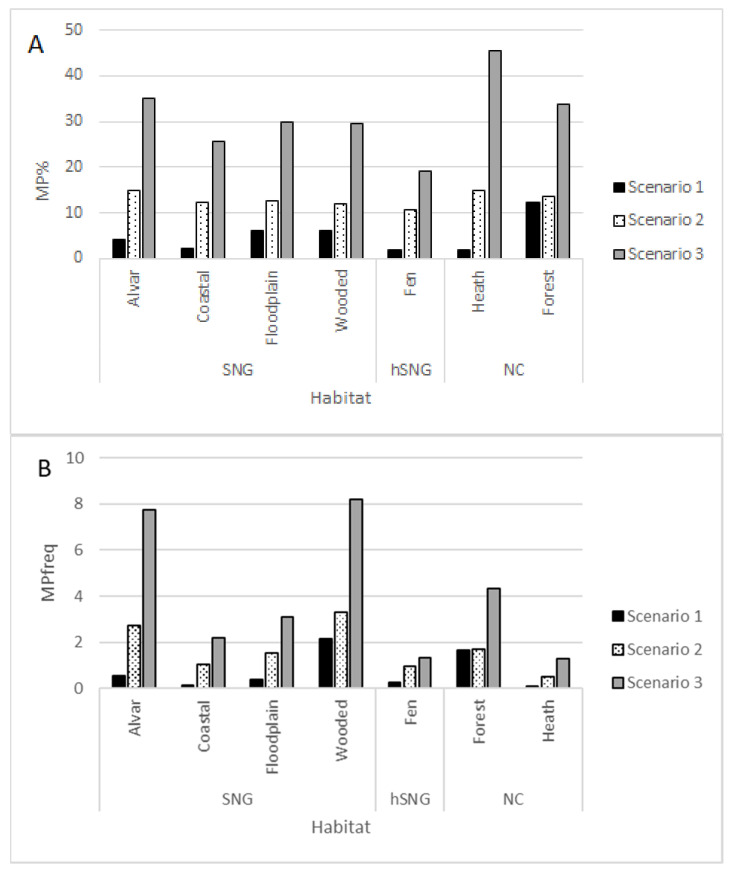
Percentage (**A**) and frequency (**B**) of MP species in Estonian habitat types according to different MP species scenarios.

**Figure 2 plants-11-00353-f002:**
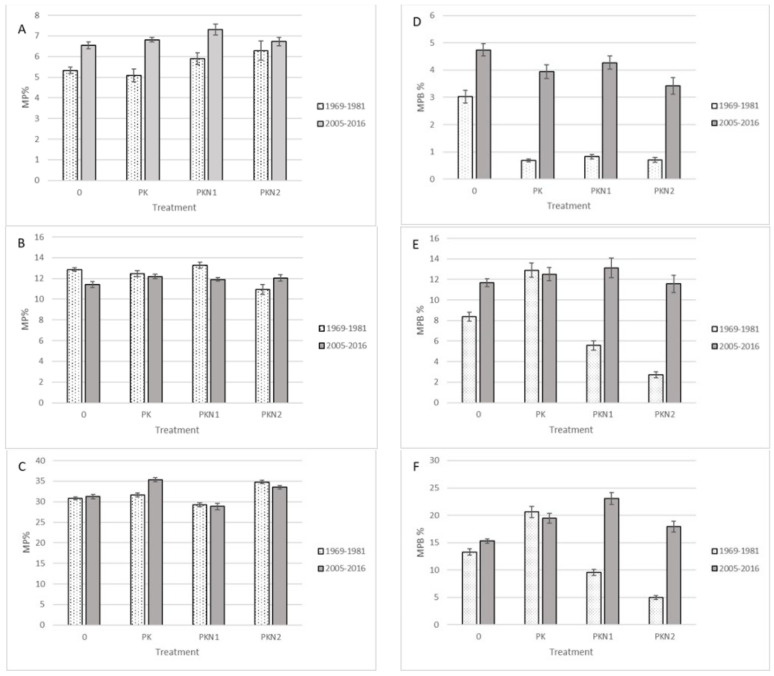
Percentage of MP species (**A**–**C**) and MP biomass (**D**–**F**) ratios in Laelatu wooded meadows during and after the fertilization experiment according to three different MP species scenarios. (**A**,**D**)—scenario 1; (**B**,**E**)—scenario 2; (**C**,**F**)—scenario 3. The vertical bars indicate standard error of average (n = 12).

**Figure 3 plants-11-00353-f003:**
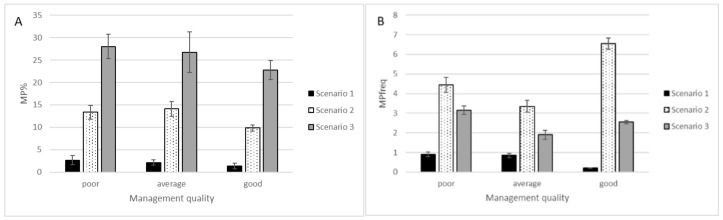
Percentage (**A**) and frequency (**B**) of MP species in plant species lists of coastal meadows with different management quality. Vertical bars indicate standard error of average (n = 3…6).

**Figure 4 plants-11-00353-f004:**
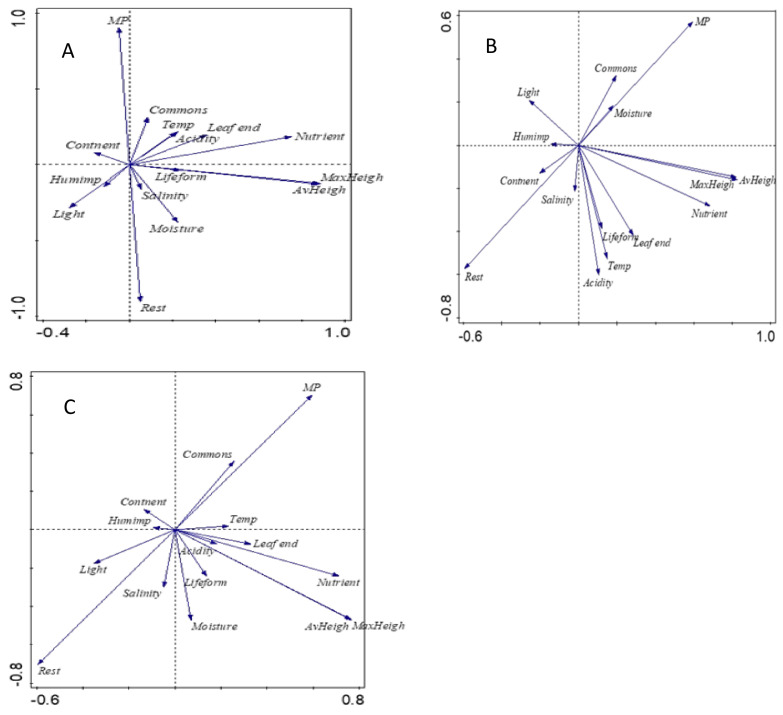
Plant requirements and characteristics of MP species compared to the rest of SNG species according to PCA test. (**A**)—scenario 1; (**B**)—scenario 2; (**C**)—scenario 3.

**Table 1 plants-11-00353-t001:** Number of MP species in various Estonian habitat types according to different scenarios.

Habitat Type	Total Species Number	MP Species		
		Scenario 1	Scenario 2	Scenario 3
Alvar	187	9	20	51
Coastal	150	8	17	40
Fen	89	2	6	16
Floodplain	194	13	22	57
Wooded	330	19	29	78
Broad-leaved forest	181	13	20	50
Heath	51	2	7	20

**Table 2 plants-11-00353-t002:** Differences between median and maximum values (%) of plant community species amount (n) according to species accumulation curves of different habitats.

Habitat Type	Community n	% of Total Species List	% of MP Species List		
			Scenario 1	Scenario 2	Scenario 3
Alvar	36	67.6	40.0	66.7	55.0
Coastal	14	59.0	50.0	36.4	54.5
Fen	12	46.7	20.0	37.5	35.7
Floodplain	8	69.0	22.2	80.0	85.7
Wooded	71	72.3	50.0	68.8	73.1
Broad-leaved forest	10	58.0	83.3	80.0	60.0
Heath	10	43.5	0.0	0.0	25.0

**Table 3 plants-11-00353-t003:** Composition of MP species list in each literature source exploited for Estonian MP species lists.

Class	Scenario 1	Scenario 2	Scenario 3
Total number of MP species in list	260	152	>1700
from these domestic	62	113	207
trees and bushes	3	25	27
fungi	3	1	3
vine	1	1	2
lichen	0	1	2
low-height shrubs and herbaceous plants	55	85	174
Main purpose of the list	Import and market control	Growing and gathering	Encyclopedic knowledge

**Table 4 plants-11-00353-t004:** Background information about the dataset used for the study. NATURA 2000 habitat codes according to [40]. Historic SNG—historically used for haymaking, currently abandoned due to wet conditions and complicated access [30].

Habitat Type	Reason	NATURA 2000 Habitat	Number of Sites	Number of Plots	Average Shannon Index
Alvars	SNG	6280* Nordic alvar and precambrian calcareous flatrocks	18	920	0.93
Coastal meadows	SNG	1630* Boreal Baltic coastal meadows	14	280	0.90
Floodplain meadows	SNG	6450 Northern boreal alluvial meadows	8	640	0.85
Fen meadows	Historic SNG	72 Calcareous fens	10	900	0.83
		90			
		7160 Fennoscandian mineral-rich springs and springfens			
Wooded meadows	SNG	6530* Fennoscandian wooded meadows	17	2800	0.91
Heath	Natural habitat	21 Sea dunes of the Atlantic, North Sea and Baltic coasts	8	800	0.76
		4030 European dry heaths			
Broad-leaved forest	Natural habitat	9010 Western taiga	8	420	0.88
		9020* Fennoscandian hemiboreal natural old broad-leaved deciduous forests (*Quercus, Tilia, Acer, Fraxinus* or *Ulmus*) rich in epiphytes			
		9050 Fennoscandian herb-rich forests with *Picea abies*			
		9060 Coniferous forests on, or connected to, glaciofluvial eskers			

**Table 5 plants-11-00353-t005:** Indexes for Ellenberg values in PCA. Av.—average of two neighbors.

Characteristic/Class	Abbreviation	1	2	3	4	5	6	7	8	9
Sensitivity to human impact	Humimp	Anthropophyte	Apophyte	Hemiradiaphore	Hemerophobe					
Commonness	Commons	Not known	Very rare	Rare	Uncommon	Scattered	Occasional	Common	Frequent	
Life form	Lifeform	Woody chamaephyte	Chamaephyte	Hemicrytophyte	Geophyte	Therophyte	Hydrophyte			
Leaf endurance	Leaf end	Evergreen	Summergreen	Springgreen						
Light	Light	Deep shade plant	Av.	Shade plant	Av.	Semi-shade plant	Av.	Plant in well-lit places	Light-loving plant	Plant in full light
Temperature	Temp	Cold, alpine	Av.	Cool, subalpine	Av.	Moderate heat indicators	Av.	Warm		
Continentality	Contnent	Euroceanic	Oceanic	Between 2 and 4	Near oceanic	Intermediate	Subcontinental	Av.	Continental	
Soil moisture	Moisture	Extreme dryness	Av.	Dry-site indicator	Av.	Moist-site indicator	Av.	Dampness indicator	Wet-site indicator	
Soil acidity	Acidity	Indicator of extreme acidity	Av.	Acidity indicator	Av.	Indicator of moderately acid soils	Av.	Weakly acid to weakly basic soils	Av.	Indicator of basic reaction
Nutrients demand	Nutrient	Indicator of extremely infertile sites	Av.	Indicator of more or less fertile soils	Av.	Intermediate fertility	Av.	Richly fertile soils	Av.	Extremely rich soils
Salinity	Salinity	Slightly salt-tolerant species	Species both in saline and non-saline	Common in coastal sites	Consistent but low salinity	Obligate halophytes	Species of mid-level saltmarsh	Species of lower saltmarsh	Permanently inundated by seawater	

## Data Availability

The original data are provided 1. Coastal meadow plant lists https://doi.org/10.15159/eds.dt.21.01 (accessed on 25 November 2020), 2. Lists of Estonian MP species by each scenario. The rest of the original data belongs to Estonian Environment Agency (https://keskkonnaagentuur.ee/en (accessed on 25 November 2020) and available on request. Appendix A downloadable also https://doi.org/10.15159/eds.art.spl.22.01 (accessed on 25 November 2020).

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
