# Peer review of "Medicinal Plants in Semi-Natural Grasslands: Impact of Management"

_plants, 2022, doi:10.3390/plants11030353_

Round 1

Reviewer 1 Report

Dear colleagues,

Your approach using the presence of medicinal plants as an indicator of ecosystem services is very interesting and innovative. Nevertheless, your paper would benefit from a major review and perhaps a change of focus.

I am not sure if the inclusion of Japanese data helps this paper, and I fear it doesn't. Please rethink and review this issue before re-submitting your manuscript.

The statistical analysis should be clarified and expanded. Obviously, you have a large group of data sets, and before estimating percentages, it would be very interesting to do species accumulation curves for each one. Then you could use a cut-off to calculate the richness of medicinal plants among the different situations.

Other things:

The abstract should be improved, explaining the methodology more clearly and including the reasons why Japan is an important part of your study. 

There are two appendices mentioned in the text (C and B), however, I did not see Appendix C. Notwithstanding, the Appendices should be sorted from A, onwards.

I include a commented pdf. Please make sure you address my comments.

Author Response

Dear reviewer,

Thank you very much for your great job and very specific feedback to our manuscript. The comments in pdf file are addressed and hopefully improved the quality of our paper.

Here are the comments to your general remarks:

Your approach using the presence of medicinal plants as an indicator of ecosystem services is very interesting and innovative. Nevertheless, your paper would benefit from a major review and perhaps a change of focus.

We agree that it is possible to write several articles about the MPs and biodiversity in the future. At the moment, however, most of the reviewers do not like MPs as there is a lack of clear scientific definition. Hence we focussed only on comparison of different habitats and management regimes in the current case.

I am not sure if the inclusion of Japanese data helps this paper, and I fear it doesn't. Please rethink and review this issue before re-submitting your manuscript.

Unfortunately our idea to make the idea of MP Ecosystem Services was probably not clear enough and therefore we excluded the Japanese part.

The statistical analysis should be clarified and expanded. Obviously, you have a large group of data sets, and before estimating percentages, it would be very interesting to do species accumulation curves for each one. Then you could use a cut-off to calculate the richness of medicinal plants among the different situations.

We agree totally that before any specific statistical analysis the data background should be studied. In our case, however the species accumulation curves reflected the reality, we knew already earlier – Estonian SNGs even In NATURA 2000 list are very heterogenic and these plant species number depend strongly on management history and/or restoration quality. Hence we did not reach to the steady state and added the original figure as supplementary material. In order to illustrate this result we added new Table 1 reflecting the diffrences between median and maximum values. Hopefully it clarifies the issue.

Other things:

The abstract should be improved, explaining the methodology more clearly and including the reasons why Japan is an important part of your study.

Japan data is excluded and abstract made more precise

There are two appendices mentioned in the text (C and B), however, I did not see Appendix C. Notwithstanding, the Appendices should be sorted from A, onwards.

Mistake corrected

Thank you for your attention and support again

On behalf of our team

Katrin Heinsoo

Reviewer 2 Report

Generally, this is well design studies to species richness in Semi-natural grasslands (SNG). Authors use medical plants as proxy to do this. The results are consistent with generally tendency.

Author Response

Dear Reviewer 2.

We thank You for the positive feedback and helpful comments. Here are our responses to the comments:

COMMENT (1) Most MP are not endangered species. Use MP as a proxy, rare and endangered species will be neglected in comparison.

ANSWER: We agree that MP-s are usually not protected or rare species in Estonia. The commonness index comes from Ellenberg Indicator value system and is an estimation for the species occurrence. It does not mean, that species is rare or endangered in the terms of nature conservation. We changed the expressions in Abstract and discussion and hopefully solved the confusion

COMMENT (2) MP change with culture and locality. The results may reflect the degree of management.

ANSWER: Yes, we do agree, that the same species might not be MP-s in other regions of Europe, or some plants, used as MP-s in one smaller locality, are not the same in another. As we removed Japanese data from the paper, the paper and interpretation has become more local and based on one cultural region only..

The results of our study confirm that management is an important issue. Also the species accumulation curves demonstrate, that in natural and nearly natural habitat the curves have reached the plateau, while in managed SNG.s there is no recognisable plateau. We added this info to our discussion

Reviewer 3 Report

The manuscript has strong deficiencies in the presentation of methods and results, which make it very difficult to understand. Before proceeding with a thorough review, it is essential that the work is presented in a clear and precise manner, therefore I invite the authors to resubmit the manuscript after this review.

Author Response

Thank You for your comments and advice.

We have addressed your comments as follows:

COMMENT: The manuscript has strong deficiencies in the presentation of methods and results, which make it very difficult to understand.

ANSWER: As there was no specification of deficiencies, we considered these to be the same, as the other reviewers outlined: we excluded the Japanese data and related information from the paper and if this was the source of difficulty we hope it is now solved.

COMMENT: Before proceeding with a thorough review, it is essential that the work is presented in a clear and precise manner, therefore I invite the authors to resubmit the manuscript after this review.

ANSWER: As there was no specification, we considered the more precise comments of other reviewers, like lack of species accumulation curves, the confusion with Japanese data etc. We have defined the species richness the better way and included evenness parameter to clarify the background of Estonian SNGs. We have made the revision and changes.

The evaluation of our English varied by reviewers and we did our best to correct our mistakes. If the poor language is still an obstacle, we are ready to ask for help from our colleague, who is a native speaker. However, then the deadline for correction should be a little longer.

Round 2

Reviewer 1 Report

I enjoyed reading the revised version of this manuscript - Medicinal plants in semi-natural grasslands: impact of management, and I now think it is acceptable for publication.

Reviewer 2 Report

 Authors answered all my concerns of manuscript. They revised the manuscript greatly. I have no more question about the mansucript.

Author Response

Dear reviewer 2. Thank you very much for your kind words about our manuscript ver 2

Reviewer 3 Report

The work deals with a little investigated and very interesting topic, offering valid ideas for further study. Nonetheless, it needs to be revised in various aspects.
Both the way it was presented and the English form require a general revision and, above all, the statistical analyses must be well defined. Below is a series of suggestions related to Results and Materials and Methods.

Results

185: “were recorded” change in “was recorded”

186: “deceasing” change in “decreasing”.

187: “study sites 50%”, add “…to 50%”.

194: what do you mean by “MP species potential”? I did not find any indication in M&M, ss that MP%? Please explain or change your sentence.

198: change “larger" to "higher" (check also elsewhere).

199: change “communities “in “habitats” (check also elsewhere).

225: “ratio of MP”, is that MP%? Please, use the same terms for the same concept.

226: “MP species availability”, see the previous note.

227: “diminutive” change in “minimal”.

248-250: it is unclear how the analysis was based in this case: please, define better.

251-252: unclear to me, however, this comment should be moved to discussion and better explained.

272-283: In my opinion, this analysis deserves to be implemented in a different way. F or Kolmogorov-Smirnov test to evaluate differences in each species traits between MP and not MP species, and Discriminant analysis (checking for assumptions), to see whether traits combination may characterise MP species.

276-278: rework this phrase.

Materials and Methods

444: Define clearly what you mean by “scenario” and the reason why analyses are performed for each scenario.

Line 481: sentence stopped without any meaning (in the downloaded file).

483-541: most of this part should be moved to 4.3.

483-484: define better this analysis and explain why you did it.

Table 4: define Shannon index in Materials and Methods

492: it is much proper to use “percentage” instead of “proportion” (if you change, check in the text)

492-495: unclear, it is better to use a general formula that can be applied to different contexts, such as:

“The percentage of MP (MP%) was calculated as the ratio between the number MP species (MPn) and the total number of plant species (PSn)”

(1)        MP%= 100*MPn /PSn

I suppose, but I am not sure, your formula was referable to habitats (it is unclear what you mean with “particular type”):

“For each habitat, MP% was calculated as the ratio between the sum of MP species number in each site (MPs) and the sum of the total number of plant species in each site (PSs)”

(2)        MP%=100*ΣMPs/ΣPSs.

However, I wonder why you did not calculate MP% of habitats as the average of the site’s MP%:

“For each habitat, MP% was calculated as the average of site’s MP%, the ratio of MP species number in each site (MPs) and the total number of plant species in each site (PSs), divided the number of sites (siten)

(2bis)   MP%=100*Σ(MPs/PSs)/siten.

506-508: unclear, which term you use? Maybe “than” instead of “thus”?

508-511: even in this case it is unclear what you mean, and it is better to use a general formula such as:

“The frequency of MP species (MPf) in each habitat was calculated as the ratio between the sum of occurrences in the plots (MPo) and the plots number (Plotn)

(3)        MPf=ΣMPo/Plotn

Hoping that I have understood.

512-521: move to a different paragraph (e.g,, 4.3 Impact of fertilization and of management).

521-523: it can be removed because included in the general formula.

523-526: being separated from the rest, change as (check is I have well understood):

“The dataset used for the Impact of fertilization allowed us to calculate the MP dry biomass ratio (MPb) as the sum of MP dry biomass weights in each plot (MPbp) divided by the total plant dry biomass in the same plot (Bp)”

(4)        MPb=Σ(MPbp/Bp).

528-537: join 512-521.

538-541: you can remove it.

544-546: remove.

546-564: explain better this part, the reason why you performed this analysis, how you prepared the matrix, the matrix traits, how you checked for PCA assumptions, the significance level of axes, and how you assessed your hypothesis.

Figures

Fig. 1: Change “Ratio” in “Percentage”

Fig. 2: - Figures A, B, and C indicates MP% (percentage), while caption indicates “MP species number … ratio”; furthermore, graphics could be better organised moving all MP% on the left, and all MPb on the right. I suggest to change in:

“Percentage (A, C, and E) and biomass ratio (B, D, and F)…. A, B – scenario 1, ….”

  • In M&M the biomass was indicated as MPb, while in fig. D, E, and F are MPb%.
  • Add scenario’s number on the figures

Fig. 3: “Average ratio..” change to “Percentage”

Fig. 4: I have some concerns about these analyses, however, I don't think the PCA shows the "impact" of plant requirements on MP. PCA shows correlations among characters.

Author Response

Dear Reviewer nr 3,

We are grateful for Your comments and suggestions that helped us to make the manuscript better. However, there was a mismatch of the line numbers of Your comments both with the line numbers of the re-submitted manuscript and with that, we downloaded for second re-submission. We did our best to improve the text and correct the mistakes. So if there is any inconvenience in our responses due to the line number mismatch, we do apologize and are willing and ready to improve the manuscript again.

Results

185: “were recorded” change in “was recorded”

done

186: “deceasing” change in “decreasing”.

done

187: “study sites 50%”, add “…to 50%”.

done

194: what do you mean by “MP species potential”? I did not find any indication in M&M, ss that MP%? Please explain or change your sentence.

Sentence changed

198: change “larger" to "higher" (check also elsewhere).

done

199: change “communities “in “habitats” (check also elsewhere).

Thank you very much, it was really our large mistake after changing the concept…

225: “ratio of MP”, is that MP%? Please, use the same terms for the same concept.

We did our best to find and replace all terms

226: “MP species availability”, see the previous note.

changed

227: “diminutive” change in “minimal”.

done

248-250: it is unclear how the analysis was based in this case: please, define better.

Due to differences in our line numbering systems we are not sure, which expression should we define better. The last sentence of the results deleted.

251-252: unclear to me, however, this comment should be moved to discussion and better explained.

See previous comment

272-283: In my opinion, this analysis deserves to be implemented in a different way. F or Kolmogorov-Smirnov test to evaluate differences in each species traits between MP and not MP species, and Discriminant analysis (checking for assumptions), to see whether traits combination may characterise MP species.

Compared to Kolmogorov-Smirnov test PCA enables to evaluate the differences between these two groups by various parameters simultaneously and to at illustrate it with ordination plots with multidimentional axes. LDA instead of PCA was an option we discussed, but in our dataset the assumption of equal class covariance and normal distribution was violated and hence we preferred to use an unsupervised method.

276-278: rework this phrase.

Again due to line number confusion we do not find the incorrect phrase

Materials and Methods

444: Define clearly what you mean by “scenario” and the reason why analyses are performed for each scenario.

Sentence changed

Line 481: sentence stopped without any meaning (in the downloaded file).

I do not see the location of the problem as in my pdf file I got from the published or in that I uploaded for re-review the line numbers are totally different, but hopefully solved the problem

483-541: most of this part should be moved to 4.3.

The following paragraphs contain info about data origin, site background and calculation methods of different parameters to describe the floristic databases without any statistical analysis in sensu stricto. Hence we think that it fits more to 4.2

483-484: define better this analysis and explain why you did it.

Your comments about PCA specification are answered later under the MM Change suggestion remarks

Table 4: define Shannon index in Materials and Methods

added

492: it is much proper to use “percentage” instead of “proportion” (if you change, check in the text)

Changed where fitted better

492-495: unclear, it is better to use a general formula that can be applied to different contexts, such as:

“The percentage of MP (MP%) was calculated as the ratio between the number MP species (MPn) and the total number of plant species (PSn)”

(1)        MP%= 100*MPn /PSn

I suppose, but I am not sure, your formula was referable to habitats (it is unclear what you mean with “particular type”):

Criteria specified

“For each habitat, MP% was calculated as the ratio between the sum of MP species number in each site (MPs) and the sum of the total number of plant species in each site (PSs)”

(2)        MP%=100*ΣMPs/ΣPSs.

done

However, I wonder why you did not calculate MP% of habitats as the average of the site’s MP%:

 “For each habitat, MP% was calculated as the average of site’s MP%, the ratio of MP species number in each site (MPs) and the total number of plant species in each site (PSs), divided the number of sites (site n)

(2bis)   MP%=100*Σ(MPs/PSs)/siten.

The database was unbalanced and therefore we wanted to be sure that our MP% takes into account the potential of the habitat (incl dark richness) that could be underestimated if smaller number of sites or if habitat includes sites with poor management (and shorter species lists)

506-508: unclear, which term you use? Maybe “than” instead of “thus”?

Changed. Unfortunately there was the shortage of info about species coverage or abundance in our databases. Therefore some other method had to be applied and some less confusing term to be used to evaluate these parameters in general

508-511: even in this case it is unclear what you mean, and it is better to use a general formula such as:

“The frequency of MP species (MPf) in each habitat was calculated as the ratio between the sum of occurrences in the plots (MPo) and the plots number (Plotn)

(3)        MPf=ΣMPo/Plotn

Hoping that I have understood.

Here is once again the problem of unbalanced database. Different sites have different number of plots and therefore site factor must be taken into account for calculations (twenty plots with MPx in site with 80 measured plots does not have same value than 20 plots with the same MP in site of 20 plots)

512-521: move to a different paragraph (e.g,, 4.3 Impact of fertilization and of management).

We are not sure if we understood you correctly. Some comments above you have suggested to include the same paragraph to 4.3 Statistical analyses. As both the Laelatu and coastal meadows databases are owned by our team and not part of database from EEA they must be described in the same subchapter

521-523: it can be removed because included in the general formula.

removed

523-526: being separated from the rest, change as (check is I have well understood):

“The dataset used for the Impact of fertilization allowed us to calculate the MP dry biomass ratio (MPb) as the sum of MP dry biomass weights in each plot (MPbp) divided by the total plant dry biomass in the same plot (Bp)”

(4)        MPb=Σ(MPbp/Bp).

I am afraid, that this is not a right formula as it summarize the ratios of different plots that is not the idea.

528-537: join 512-521.

538-541: you can remove it.

544-546: remove.

These three suggestions are difficult to follow as the line numbers in any of our versions does not correspond with these in your pdf, but we tried our best

546-564: explain better this part, the reason why you performed this analysis, how you prepared the matrix, the matrix traits, how you checked for PCA assumptions, the significance level of axes, and how you assessed your hypothesis.

PCA method is widely used in plant ecology. The search in Web of Science gives more than 300 publications on plant ecology & Principal Component Test and hence we are not sure we have to explain it in details. For instance, Dray&Josse (2017) state that “Principal component analysis (PCA) is a standard technique to summarize the main structures of a data table containing the measurements of several quantitative variables for a number of individuals.” The parameters used for matrix generation (traits?) and these quantification values are described in Table 5. The cumulative signification level of first two axes reported in Results “despite the scenario the first two axes did not explain more than 35% of the variation”. PCA does not give any quantitative value for the hypothesis assessment, but demonstrates only trends through ordination result visualization. Therefore we tried to formulate our results accordingly. 

Fig. 1: Change “Ratio” in “Percentage”

done

Fig. 2: - Figures A, B, and C indicates MP% (percentage), while caption indicates “MP species number … ratio”; furthermore, graphics could be better organised moving all MP% on the left, and all MPb on the right. I suggest to change in:

Caption changed and the location of the figures moved.

“Percentage (A, C, and E) and biomass ratio (B, D, and F)…. A, B – scenario 1, ….”

We are afraid this version can be confusing…

  • In M&M the biomass was indicated as MPb, while in fig. D, E, and F are MPb%.

We changed it in M&M

  • Add scenario’s number on the figures
  • We do not want to double it with legend. Hopefully with the changed figure location it is clear now

Fig. 3: “Average ratio..” change to “Percentage”

changed

Fig. 4: I have some concerns about these analyses, however, I don't think the PCA shows the "impact" of plant requirements on MP. PCA shows correlations among characters.

Sentence changed

Round 3

Reviewer 3 Report

The authors have accepted several suggestions, largely improving the manuscript; however, there are some important questions that must be resolved. In particular, the formulas they used have a significant fault. Furthermore, I still question the use of a PCA. Below are my comments to the Authors replies (other comments are in the attached file).

483-541: most of this part should be moved to 4.3.

The following paragraphs contain info about data origin, site background and calculation methods of different parameters to describe the floristic databases without any statistical analysis in sensu stricto. Hence we think that it fits more to 4.2

  • Calculation methods relate to statistical analysis, I suggest moving all these parts to 4.3, leaving in 4.2 only information about data collection.

492-495: unclear ….

However, I wonder why you did not calculate MP% of habitats as the average of the site’s MP%:

“For each habitat, MP% was calculated as the average of site’s MP%, the ratio of MP species number in each site (MPs) and the total number of plant species in each site (PSs), divided the number of sites (site n)

(2bis) MP%=100*Σ(MPs/PSs)/siten.

The database was unbalanced and therefore we wanted to be sure that our MP% takes into account the potential of the habitat (incl dark richness) that could be underestimated if smaller number of sites or if habitat includes sites with poor management (and shorter species lists)

  • The formula that you proposed has a main fault, because it doesn’t calculate the average MP% but a value that is dependent on the species number size. That is, habitats with a lower number of species have a higher MP%, even if the percentage of MP should be the same.

508-511: even in this case it is unclear what you mean, and it is better to use a general formula such as:

“The frequency of MP species (MPf) in each habitat was calculated as the ratio between the sum of occurrences in the plots (MPo) and the plots number (Plotn)

(3) MPf=ΣMPo/Plotn

Hoping that I have understood.

Here is once again the problem of unbalanced database. Different sites have different number of plots and therefore site factor must be taken into account for calculations (twenty plots with MPx in site with 80 measured plots does not have same value than 20 plots with the same MP in site of 20 plots)

  • Your comment is unclear to me, a frequency is not dependent on the number of cases; you have to calculate the frequency of each species (MPf, see formula n. 3) and then the average frequency for the habitat: MPfreq=ΣMPf/number MP

523-526: being separated from the rest, change as (check is I have well understood):

“The dataset used for the Impact of fertilization allowed us to calculate the MP dry biomass ratio (MPb) as the sum of MP dry biomass weights in each plot (MPbp) divided by the total plant dry biomass in the same plot (Bp)”

(4) MPb=Σ(MPbp/Bp).

I am afraid, that this is not a right formula as it summarize the ratios of different plots that is not the idea.

  • Yes, you are right, I have forgotten a part: MPb=Σ(MPbp/Bp)/number of plots. However, your formula has the same fault as in previous formulas.

546-564: explain better this part, the reason why you performed this analysis, how you prepared the matrix, the matrix traits, how you checked for PCA assumptions, the significance level of axes, and how you assessed your hypothesis.

PCA method is widely used in plant ecology. The search in Web of Science gives more than 300 publications on plant ecology & Principal Component Test and hence we are not sure we have to explain it in details. For instance, Dray&Josse (2017) state that “Principal component analysis (PCA) is a standard technique to summarize the main structures of a data table containing the measurements of several quantitative variables for a number of individuals.” The parameters used for matrix generation (traits?) and these quantification values are described in Table 5. The cumulative signification level of first two axes reported in Results “despite the scenario the first two axes did not explain more than 35% of the variation”. PCA does not give any quantitative value for the hypothesis assessment, but demonstrates only trends through ordination result visualization. Therefore we tried to formulate our results accordingly.

  • Thank you for the explanation, however, I have some questions:
    • PCA assumes continuous variables and multivariate normality, or at least ordinal variables with many ordinal steps, a reduced skewness, few double zeros, and no significant outliers: did you check for them?
    • There needs to be a linear relationship between all variables, did you check?
    • Multicollinearity must be avoided: did you check?
    • Data need to be standardised, how did you do it?
    • You should have sampling adequacy, how large was your matrix?
    • Are your data suitable for reduction? (i.e. Barlett’s test)
    • Which are the eigenvalues of the first two axes? Which is the percentage of the explaned variance of each?

272-283: In my opinion, this analysis deserves to be implemented in a different way. F or Kolmogorov-Smirnov test to evaluate differences in each species traits between MP and not MP species, and Discriminant analysis (checking for assumptions), to see whether traits combination may characterise MP species.

Compared to Kolmogorov-Smirnov test PCA enables to evaluate the differences between these two groups by various parameters simultaneously and to at illustrate it with ordination plots with multidimentional axes. LDA instead of PCA was an option we discussed, but in our dataset the assumption of equal class covariance and normal distribution was violated and hence we preferred to use an unsupervised method.

  • I moved down here this point because it is related to the previous comment. I think PCA is not suitable for your data, as well as LDA. To check relationships in a multidimensional space with this kind of data, Correspondence Analysis is the best method, followed by a non-parametric test (I suggested Kolmogorov-Smirnov test) to check the significance of observed correlated traits.

Author Response

Dear Reviewer 3,

We are really thankful for such patient and detailed review and we discussed all the problematic issues with all the co-authors plus searched for additional literature sources. Our further new comments are in Bold+Italic

The authors have accepted several suggestions, largely improving the manuscript; however, there are some important questions that must be resolved. In particular, the formulas they used have a significant fault. Furthermore, I still question the use of a PCA. Below are my comments to the Authors replies (other comments are in the attached file).

483-541: most of this part should be moved to 4.3.

The following paragraphs contain info about data origin, site background and calculation methods of different parameters to describe the floristic databases without any statistical analysis in sensu stricto. Hence we think that it fits more to 4.2

  • Calculation methods relate to statistical analysis, I suggest moving all these parts to 4.3, leaving in 4.2 only information about data collection.
  • We specified the headline of 4.2. Moving the equations to statistical analyses subchapter would lead the reader to opinion that we will run some statistic’s tool on these data.

 492-495: unclear ….

However, I wonder why you did not calculate MP% of habitats as the average of the site’s MP%:

“For each habitat, MP% was calculated as the average of site’s MP%, the ratio of MP species number in each site (MPs) and the total number of plant species in each site (PSs), divided the number of sites (site n)

(2bis) MP%=100*Σ(MPs/PSs)/siten.

The database was unbalanced and therefore we wanted to be sure that our MP% takes into account the potential of the habitat (incl dark richness) that could be underestimated if smaller number of sites or if habitat includes sites with poor management (and shorter species lists)

  • The formula that you proposed has a main fault, because it doesn’t calculate the average MP% but a value that is dependent on the species number size. That is, habitats with a lower number of species have a higher MP%, even if the percentage of MP should be the same.

I am afraid we have to disagree here. Both the formula that we used and that you suggest are not depending on the number of plant species per habitat (see incl table) and the main difference in these is the impact of sites with less species. The reason, why we wanted to avoid this phenomenon, is described in our previous answer

low species case

high species case

MP

PS

MP/PS

MP

PS

MP/PS

1

5

0.2

3

15

0.2

1

2

0.5

3

6

0.5

2

7

0.285714

6

21

0.285714

3

9

0.333333

9

27

0.333333

sums

7

23

21

69

sum(MP)/sum(PS

0.304348

0.329762

0.304348

0.329762

508-511: even in this case it is unclear what you mean, and it is better to use a general formula such as:

“The frequency of MP species (MPf) in each habitat was calculated as the ratio between the sum of occurrences in the plots (MPo) and the plots number (Plotn)

(3) MPf=ΣMPo/Plotn

Hoping that I have understood.

Here is once again the problem of unbalanced database. Different sites have different number of plots and therefore site factor must be taken into account for calculations (twenty plots with MPx in site with 80 measured plots does not have same value than 20 plots with the same MP in site of 20 plots)

  • Your comment is unclear to me, a frequency is not dependent on the number of cases; you have to calculate the frequency of each species (MPf, see formula n. 3) and then the average frequency for the habitat: MPfreq=ΣMPf/number MP

I am afraid I did not get your last idea – maybe something is missing in your last formula? Your previous suggestion to make this formula more general, will be lost in too sophisticated equation and we prefer our formula as easier to understand and in better harmonization with eq 1.

523-526: being separated from the rest, change as (check is I have well understood):

“The dataset used for the Impact of fertilization allowed us to calculate the MP dry biomass ratio (MPb) as the sum of MP dry biomass weights in each plot (MPbp) divided by the total plant dry biomass in the same plot (Bp)”

(4) MPb=Σ(MPbp/Bp).

I am afraid, that this is not a right formula as it summarize the ratios of different plots that is not the idea.

  • Yes, you are right, I have forgotten a part: MPb=Σ(MPbp/Bp)/number of plots. However, your formula has the same fault as in previous formulas.
  • The indicated phenomenon is described with an illustrative model above

546-564: explain better this part, the reason why you performed this analysis, how you prepared the matrix, the matrix traits, how you checked for PCA assumptions, the significance level of axes, and how you assessed your hypothesis.

PCA method is widely used in plant ecology. The search in Web of Science gives more than 300 publications on plant ecology & Principal Component Test and hence we are not sure we have to explain it in details. For instance, Dray&Josse (2017) state that “Principal component analysis (PCA) is a standard technique to summarize the main structures of a data table containing the measurements of several quantitative variables for a number of individuals.” The parameters used for matrix generation (traits?) and these quantification values are described in Table 5. The cumulative signification level of first two axes reported in Results “despite the scenario the first two axes did not explain more than 35% of the variation”. PCA does not give any quantitative value for the hypothesis assessment, but demonstrates only trends through ordination result visualization. Therefore we tried to formulate our results accordingly.

  • Thank you for the explanation, however, I have some questions:
    • PCA assumes continuous variables and multivariate normality, or at least ordinal variables with many ordinal steps, a reduced skewness, few double zeros, and no significant outliers: did you check for them?
    • There needs to be a linear relationship between all variables, did you check?
    • Multicollinearity must be avoided: did you check?
    • Data need to be standardised, how did you do it?
    • You should have sampling adequacy, how large was your matrix?
    • Are your data suitable for reduction? (i.e. Barlett’s test)
    • Which are the eigenvalues of the first two axes? Which is the percentage of the explaned variance of each?

272-283: In my opinion, this analysis deserves to be implemented in a different way. F or Kolmogorov-Smirnov test to evaluate differences in each species traits between MP and not MP species, and Discriminant analysis (checking for assumptions), to see whether traits combination may characterise MP species.

Compared to Kolmogorov-Smirnov test PCA enables to evaluate the differences between these two groups by various parameters simultaneously and to at illustrate it with ordination plots with multidimentional axes. LDA instead of PCA was an option we discussed, but in our dataset the assumption of equal class covariance and normal distribution was violated and hence we preferred to use an unsupervised method.

  • I moved down here this point because it is related to the previous comment. I think PCA is not suitable for your data, as well as LDA. To check relationships in a multidimensional space with this kind of data, Correspondence Analysis is the best method, followed by a non-parametric test (I suggested Kolmogorov-Smirnov test) to check the significance of observed correlated traits.

We want to address, that we used the PCA analyse only for analysing the dataset, created from Ellenberg indicator values (EIV) to indicate any ecological features, that might be more common to MP-s than to other species in our dataset. In other words, all species got the EIV-s and in PCA we just analysed these ecological parameters (For details see Table 5) plus heights. We assumed that well-known and much-cited EIVs are theoretically independent from each other and any multicolinnearity study will lead to data-hunting results only. We had no reasonable assumptions to relate these parameters with the height ones as well. Standardisation procedure and excluding of all rows with empty cells are default steps in PCA. Therefore the matrix was only 13 parameters ((rows 459-460) * 171 species (stated in row 212).  The cumulative eigenvalue of first two axes in our models were ca 35% (stated in row 212).

We are aware that usage of PCA and all tools with the same technique is always disputable and statements based on these models should be modest and we tried our best to keep our text in this style. This kind of approach, however, has been used by Pärtel et al (1999) for alvars, Horsak et al (2007) for plant communities in association with snails, Tintner and Klug (2011) for landfill cover vegetation, Odland (2009) for vegetation altitudinal gradients in Norway, Hedl (2004) for assessing changes in Beech forests and many others. On the other hand, we did not find any matches for Ellenberg values and Kolmogorov-Smirnov test in Web of Science database and hence we are afraid that the results that we should publish with this method will be less understandable to the readers. Therefore we are still on opinion, that our approach to analyse EIV with PCA test is relevant and correspond to similar studies that we have read about from published and well-cited articles.

Pärtel, M., Kalamees, R., Zobel, M, & Rosén, E. 1999. Alvar grasslands in Estonia: variation in species composition and community structure. Journal of Vegetation Science 10: 561-570

Horsak, M., Hajeka, M., Tichy, L., Jurickova, L. 2007. Plant indicator values as a tool for land mollusc autecology assessment. Acta Oecologica 32:161 – 171

Tintnera,J., Klug, B. 2011 Can vegetation indicate landfill cover features? Flora 206:559–566

Hedl, R. 2004. Vegetation of beech forests in the Rychlebské Mountains, Czech Republic, re-inspected after 60 years with assessment of environmental changes. Plant Ecology 170: 243–265

Odland, A. 2009.  Interpretation of altitudinal gradients in South Central Norway based on vascular plants as environmental indicators.  Ecological Indicators 9: 409 – 421
